

# Prevalence of *Theileria ovis* in sheep and goats in northwestern Saudi Arabia with notes on potential vectors

Ruoa S. Almahallawi[1], Sawsan A. Omer[2], Esam M. Al-Shaebi[2], Nawal Al-Hoshani[3], Esam S. Al-Malki[4], Rewaida Abdel-Gaber[2] and Osama B. Mohammed[2]

[1] Department of Biology, University College of Duba, Tabuk University, Tabuk, Saudi Arabia
[2] Department of Zoology, College of Science, King Saud University, Riyadh, Saudi Arabia
[3] Department of Biology, College of Science, Princess Nourah bint Abdulrahman University, Riyadh, Saudi Arabia
[4] Department of Biology, College of Science in Zulfi, Majmaah University, Majmaah, Saudi Arabia

Corresponding author
Osama B. Mohammed,
obmkkwrc@yahoo.co.uk

## ABSTRACT

The prevalence of *Theileria* spp. infecting sheep and goats were investigated in two cities and surroundings in northwest Saudi Arabia. Blood samples from 403 sheep and goats from Madina ($n = 201$) and Tabuk ($n = 202$) cities (177 from sheep and 226 from goats) were investigated. Blood samples were examined microscopically for the presence of intraerythrocytic bodies suggestive of *Theileria* as well as they were investigated using molecular techniques. DNA was extracted from blood and ticks and subjected to polymerase chain reaction amplification using specific primers. The primers used amplified a fragment of the 18S rRNA region (450 bp) targeting the hypervariable region IV. A total of 63 ticks belonging to five species were collected from sheep and goats for determination of their involvement of lifecycle of *Theileria*. Ticks were identified morphologically and confirmed molecularly utilizing cytochrome oxidase C subunit 1 gene (COXI) amplification. The results indicated that: microscopic examination revealed 24 (6%) of the samples investigated showed intraerythrocytic bodies suggestive of *Theileria*. Positive samples were only obtained from sheep whereas goats yielded negative results. A total of 33 (18.6%) sheep samples were positive for *Theileria* infection using polymerase chain reaction targeting the partial 18S rDNA and DNA sequencing. *Theileria* infection was more prevalent in animals that were less than 2 years of age compared with older animals. There was no difference in the prevalence of the infection between male and female sheep in both cities. All positive sheep were detected during the summer and none of the samples collected during the winter were positive. Phylogenetic analysis revealed that the sequences obtained from *Theileria* species reported in the present study grouped with sequences from *Theileria ovis* from different countries. Ticks were identified as *Hyalomma dromedarii*, *Hyalomma, marginatum*, *Hyalomma impeltatum* and *Hyalomma anatolicum anatolicum*. *T. ovis* DNA was detected from *Hyalomma dromedarii* and *Hyalomma impeltatum* suggesting that they are potential vectors of this piroplasm in sheep from Madina and Tabuk cities. This report is considered the first report of *T. ovis* infecting sheep from Madina and Tabuk, furthermore, it is the first report determining the vectors responsible for transmission of the infection in sheep in northwest Saudi Arabia. The data generated from this

study will undoubtedly pave the way for the detection and control of ovine and caprine theileriosis in Madina and Tabuk regions.

## INTRODUCTION

Sheep and goats may be infected by several pathogens with a wide variety of parasites among which the gastrointestinal parasitic infection is the most common and these include helminthic infections, protozoan diseases including coccidiosis in addition to haemoparasites (*Ngole, Ndamukong & Mbuh, 2001*; *Okaiyeto et al., 2008*; *Estrada-Reyes et al., 2019*). Haemoparasites such as the protozoan parasites; *Theileria, Babesia, Trypanosoma*, as well as the bacteria; *Rickettsia, Anaplasma*, and *Ehrlichia* (*Cowdria*), are widely spread among ruminants. The three protozoans listed above are the most prevalent haemo protozoa in sheep and goats. These protozoa are transmitted through arthropod vectors and have generally been shown to cause lysis of red blood corpuscles resulting in anemia, anorexia, high morbidity and mortality, infertility, jaundice, and weight loss (*Mehlhorn, Schein & Ahmed, 1994*; *Okaiyeto et al., 2008*; *Ademola & Onyiche, 2013*; *Sharifi et al., 2016*). Among *Theileria* species affecting small ruminants, *Theileria lestoquardi* (causing malignant ovine theileriosis), *T. uilenbergi* and *T. luwenshuni* are considered highly pathogenic, while other *Theileria* species are rarely or entirely non-pathogenic such as; *T. ovis* (agent of benign theileriosis), *T. recondita* (causing mild ovine theileriosis) (*Torina & Caracappa, 2012*; *Stuen, 2020*).

In Saudi Arabia, theileriosis in sheep was studied in the central part and a restricted part of the western region (*Alanazi et al., 2019*; *Metwally et al., 2021*). A prevalence of 57.8% of *Theileria* infection was reported from sheep and 51.9% from goats in western Saudi Arabia using molecular methods (*Metwally et al., 2021*). *Alanazi et al. (2019)* reported a prevalence of 33.2% from sheep and 25.2% from goats as well as 46% from sheep and 33.7% from goats using microscopic examination and molecular techniques respectively. *Ashraf et al. (2024)* reported a low prevalence of 3.3% of *T. lestoquardi* from sheep and not from goats from Dera Ghazi Khan district, Pakistan. Molecular prevalence of *T. ovis*, in the same country, using the *18S rRNA* gene was also investigated by *Arif et al. (2023)*. Both sheep and goats were found to infected at low rates, 4% and 2% respectively.

According to the national gross domestic product (GDP), livestock contributes about 40% of the global value of agricultural outputs, wires the livelihood and food security of almost 1.3 billion people, and increases opportunities for agricultural development, poverty reduction, and food security globally (*Food and Agriculture Organization of the United Nations (FAO), 2009*). Small ruminants (sheep and goats) account for the main livestock, as they contribute significantly to the meat, milk, skin production, and socio-cultural values to human development because of food provision and the overall economy of certain nations raising such animals (*Ajayi & Ajayi, 1983*; *Sebsibe, 2008*;
*Adamu & Balarabe, 2012*; *Sargison, 2020*). Furthermore, many studies suggest that the production of small ruminants is a major source of food and non-food products, thus generating more market opportunities and an integral part of the nomadic culture of the Middle East including Saudi Arabia (*Abouheif et al., 1989*; *Jaber, Diehl & Hamadeh, 2016*; *Rehman et al., 2017*; *Salami et al., 2019*).

There is a paucity of information on the haemoparasites of sheep and goats and their significance on health and productivity as well as potential vectors which might be involved in their life cycles. Recently, molecular tools have increasingly become an integral part of studying the epidemiology of infectious diseases for specific identification and the phylogenetic relationship of different disease-causing agents. Detailed information concerning haematozoan vectors, carriers, and reservoir hosts is essential if parasite control measures are to be implemented (*Torina et al., 2007*). This study reports an epidemiological survey of *Theileria* parasites in sheep and goats in northern western Saudi Arabia as well as evaluating the associated risk factors. Previous work from the area utilized microscopic methods and only provisional data were obtained. Molecular studies will undoubtedly detect the strain of the parasites existing in the area and also determining the possible vector involved in the lifecycle will help in the control of the *Theileria* spp. involved.

In the present study, the prevalence of *Theileria* parasites was investigated on both sheep and goats from two cities in the northwest of Saudi Arabia. The potential vector responsible of the transmission of this haemoparasite from sheep and goats was also identified.

## MATERIALS AND METHODS

The King Saud University ethics approval to carry out the study with the Department of Zoology Facility (Ethics Reference: KSU-SE.23-104).

### Animals and samples

The present study was conducted between 2020 to June 2021 in the vicinity of two cities in northwestern Saudi Arabia (Madina (24.8404° N, 39.3206° E) and Tabuk (28.2453° N, 37.6387° E)) (Fig. 1). Madina is located in the Hejaz region of Western Saudi Arabia about 340 km north of Makkah. The city is at an elevation of approximately 620 m (2,030 feet) above sea level. The rest of the area is occupied by the Hejaz mountain range, empty valleys, agricultural spaces, older dormant volcanoes and the Nafud desert. Madina is located in a hot desert climate region. In summer, the highest average temperature ranges between 40–43 °C with nights about 29 °C. Winters are milder, with temperatures from 12 °C at night to 25 °C in the day. There is very little rainfall, which falls almost entirely between November and May. Humidity in Madina is ~35 between November and March whereas it is 15–20 between April and October.

Tabuk city is located at the north-west coast of the country, facing Egypt across the Red Sea, it is close to the Jordanian and Saudi Arabian borders. Tabuk city is at an elevation of 768 m (2,520 ft) above sea level. Tabuk's climate is a desert one. The average temperature

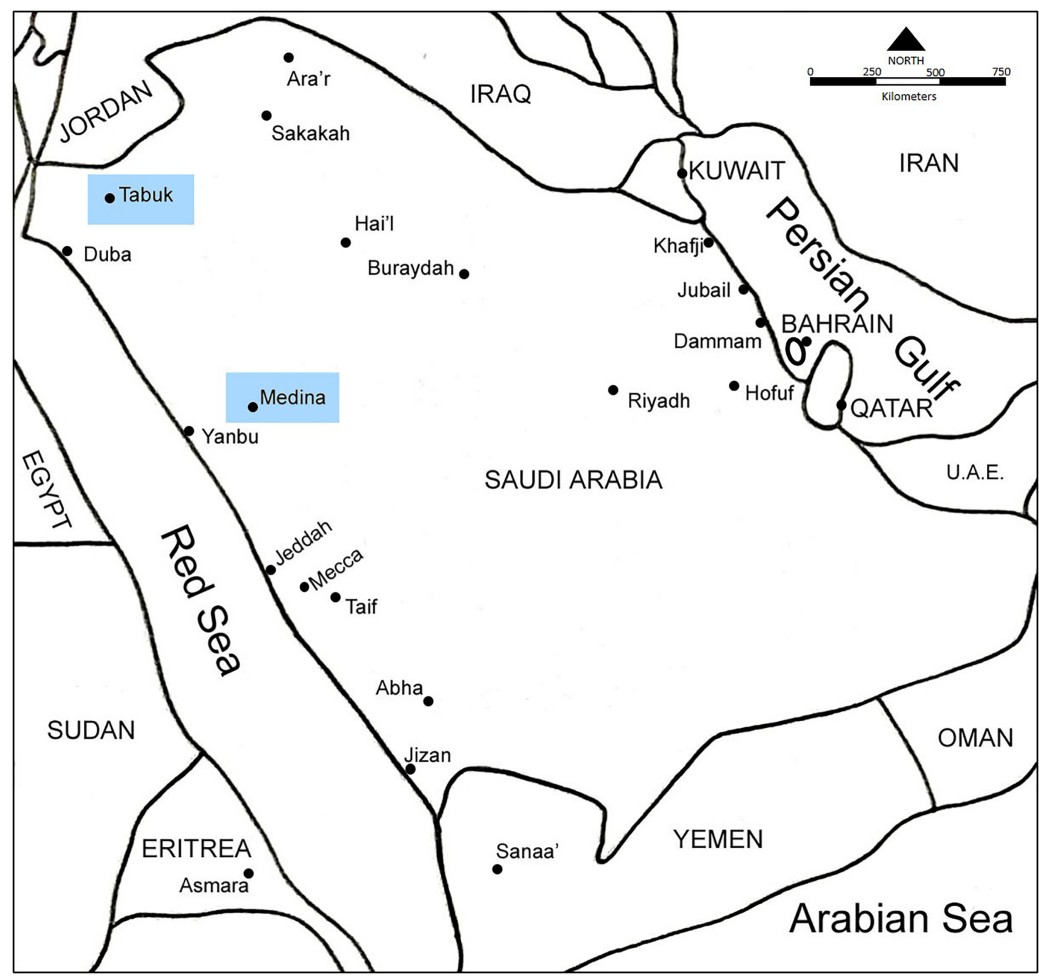

**Figure 1 A map showing Saudi Arabia and the location of the two cities (Madina and Tabuk) where the current study was conducted.**

in Tabuk is 21.6 °C. Precipitation is around 46 mm 1.8 inch per year. Humidity in Tabuk is ~50 between November and March whereas it is 20–30 between April and October.

The sample size ($n$) was determined using the statistical formula by *Thrusfield (2018)*:

$$n = \frac{Z^2 \, P(1-P)}{d^2}$$

where $n$ is the desired sample size; $Z$ is 95% confidence interval (1.96), $P$ is the estimated prevalence of infections and $d$ is the desired precision level (5%). Since the prevalence of *Theileria* infections in the study area was unknown and this study was undertaken to determine the prevalence of *Theileria* spp. infections in sheep and goats, 15% prevalence was assumed (*Thrusfield, 2018*).

A total of 403 blood samples were collected from sheep ($n = 177$) and goats ($n = 226$). A total of 201 (77 sheep and 124 goats were collected from Madina) whereas, 202 (100 sheep and 102 goats) samples were collected from Tabuk. All blood samples were collected from

the jugular vein of animals in 5 ml tubes with EDTA as an anticoagulant. The samples were collected during the summer ($n$ = 219) and winter ($n$ = 184).

Sheep and goats' age, gender, location, rearing system, health status, and tick infestation were observed and recorded for each individual used in the present study.

Animals from which blood samples were collected were kept either under intensive rearing or under open systems. Under intensive rearing system, food and water are provided while they in holding pens. Whereas under open systems animals will be allowed to wonder around during the day and then the are housed in small stall at night. Shepherds are always taking care of animals and making sure food and water are available and animals are not exposed to extreme environmental conditions. Animals were handled gently by shepherd during the short time of the samples collection. No analgesia was used during the samples collection as the process was quick and the veterinarian who is taking the blood is experienced. Animals are set free after sample collection.

Methanol-fixed blood smears stained with 5% Giemsa stain were prepared from each blood sample and examined microscopically for demonstration of intraerythrocytic blood parasites.

Sixty-three ticks were collected from sheep and goats from which blood samples were collected. Ticks were preserved in 70% ethanol and identified to the species level according to the keys given by *Hoogstraal, Wassef & Büttiker (1981)*, *Diab et al. (1987)*, *Walker (2003)*, *Apanaskevich & Horak (2008)*. All stages of ticks were adults and 30 ticks were collected from goats (12 from Madina; 18 from Tabuk) and 33 from sheep (19 from Madina; 14 from Tabuk).

## DNA extraction, PCR amplification, sequencing, and phylogenetic analysis

DNA from all the blood samples was extracted using the DNeasy Blood & Tissue Kit spin-column protocol Qiagen (Hilden, Germany) following the manufacturer's protocol. Extracted DNA was stored at −20 °C until use. Ticks were washed individually in 500 ml of sterile double distilled water to remove the ethanol and crushed individually using Liquid nitrogen, in a sterile ceramic mortar and tissue homogenizer, placed in a sterile 1.5 ml Eppendorf tube. Then the samples were treated as blood samples. The purified genomic DNA was recovered and stored at −20 °C until used.

A fragment of ~450-base pair (bp) of the 18S small subunit ribosomal RNA (rRNA) gene was amplified using the primers RLB-F [5′- GACACAGGGAGGTAGTGACAAG-3′] and RLB-R [5′- CTAAGAATTTCACCTCTGACAGT-3′], which detect the DNA of both *Theileria* and *Babesia* species (*Georges et al., 2001*). The polymerase chain reaction (PCR) was set up using 2 µl of the DNA extracted from blood with 10 µM of each of the forward and reverse primers, 5 µl of 5× PCR buffer, and 0.2 µl (1 unit) of Taq DNA polymerase (Bioline, London, UK). Double distilled water was used as the negative control to detect contamination. The PCR was performed on a thermocycler (Multigene; Labnet, Edison, NJ, USA) and the conditions for the PCR were as follows: 1 initial denaturation cycle at 94 °C for 2 min, followed by 35 cycles at 94 °C for 30 s, 57 °C for 30 s, and 72 °C for 30 s, and a final extension at 72 °C for 5 min. Primers that were used for the amplification

of cytochrome oxidase C subunit 1 gene (COXI) from ticks were COXI-F 5′-GGAACAA TATATTTAATTTTTGG-3′ and COXI-R 5′-ATCTATCCCTACTGTAAATATATG-3′ which amplify ~800 bp of the COXI (*Chitimia et al., 2010*).

The PCR products were checked using 1.5% agarose gel electrophoresis and visualized with ultraviolet light using a transilluminator. Digital images were taken of the PCRs.

The PCR products were sequenced following the standard procedure of Sanger sequencing at Macrogen, South Korea. All sequences were aligned analyzed and edited using MEGA X software (*Tamura, Stecher & Kumar, 2021*). Further, the phylogenetic trees were based on the comparison with sequences of *Theileria* spp. detected in the present study and those available in GenBank. Trees were constructed using the maximum likelihood (ML) and neighbor-joining (NJ) methods of the MEGA X program and bootstrap analysis with 1,000 replications was used to estimate the confidence of the branching patterns of the trees (*Tamura, Stecher & Kumar, 2021*). The evolutionary history inferred by the maximum likelihood method was based on the Tamura-Nei model (*Tamura & Nei, 1993*).

## Statistical analysis

Differences in the prevalence rates of *Theileria* spp. in both cities from sheep and goats as well as from males, females, locality, rearing system, and different seasons were tested using $\chi 2$ -test, which was performed in SPSS 18.0. The significance was set at 95%. The sensitivity of the PCR method compared to the microscopic method was estimated according to formula outlined by *Noaman (2014)*.

# RESULTS

## Microscopic examination

The prevalence of *Theileria* infection using both microscopic and PCR methods is given in Table 1. Microscopic examination of Giemsa-stained blood smears from sheep revealed that intraerythrocytic stages suggestive of *Theileria* spp. were detectable from sheep and not from the goats investigated (Fig. 2). *Theileria* spp. were detected in 24 (13.6%) sheep samples; 14 (18.2%) were from Madina and 10 (10%) were from Tabuk (Table 1). There was no significant difference in the prevalence of *Theileria* spp. between sheep in Madina and Tabuk using the microscopic method ($X^2$ = 0.3528, $p$ > 0.05). Microscopically, there was no significant difference in the prevalence of *Theileria* spp. in sheep based on the sex, age, and rearing system from both cities. All positive samples were reported during the summer and none collected during the winter was positive. Despite the fact that more females, from both cities, are positive to *Theileria* infection, using microscopic method, however, the difference was not significant ($p$ > 0.05). Even in Tabuk where seven females were positive and three males were positive, the difference was not significant (Table 1).

## Ticks identification

Ticks from the sheep and goats in Madina and Tabuk were collected to rule out the specific vector for *Theileria* spp. transmission. A total of sixty-three ticks were collected during the study period, 33 were from sheep whereas 30 were from goats. They were identified

**Table 1 Results of microscopic and PCR investigations showing the prevalence of *Theileria ovis* infection in sheep from Madina and Tabuk cities.** The results of different risk factors studied is also shown.

| Variables | Number examined | | Microscopic results (%) | | *P* value | PCR Results (%) | | *P* value |
|---|---|---|---|---|---|---|---|---|
| | Madina | Tabuk | Madina | Tabuk | | Madina | Tabuk | |
| **Sex** | | | | | >0.05 | | | >0.05 |
| Male | 36 | 48 | 7 (19.4) | 3 (6) | | 11 (30) | 5 (10) | |
| Female | 41 | 52 | 7 (17) | 7 (3.4) | | 9 (22) | 8 (15.3) | |
| **Season** | | | | | - | | | - |
| Summer | 57 | 55 | 14 (24.5) | 10 (18) | | 20 (35) | 13 (23.6) | |
| Winter | 20 | 45 | 0 | 0 | | 0 | 0 | |
| **Age** | | | | | >0.05 | | | <0.05 |
| <2 Years | 16 | 43 | 7 (44) | 4 (9.3) | | 14 (87.5) | 6 (14) | |
| >2 Years | 61 | 57 | 7 (11.5) | 6 (10.5) | | 6 (9.8) | 7 (12.2) | |
| **Rearing system** | | | | | >0.05 | | | >0.05 |
| Open | 42 | 51 | 5 (12) | 7 (14) | | 9 (21.4) | 10 (20) | |
| Intensive | 35 | 49 | 9 (25.7) | 3 (6.1) | | 11 (31.4) | 3 (6.1) | |

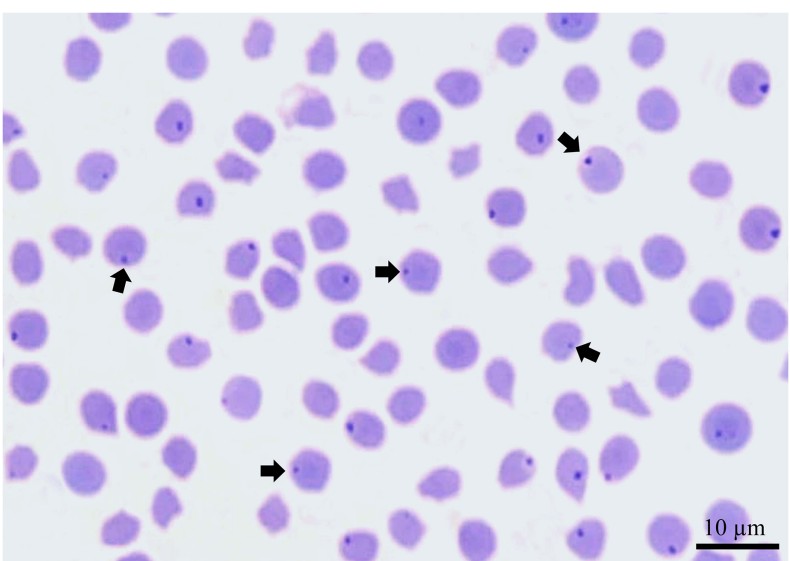

**Figure 2  A Giemsa-stained blood smear from a sheep showing intraerythrocytic bodies suggestive of *Theileria* (arrows).** Scale bar: 10 µm. 

morphologically as belonging to the genus *Hyalomma*. *H. impeltatum* (46), *H. dromedarii* (14), *H. anatolicum anatolicum* (2) and *H. marginatum* (1). DNA sequences obtained from representative individuals confirmed the identity of morphological description. DNA sequences from ticks related to COXI were deposited in GenBank with the accession numbers (ON138769–ON138791) as *H. impeltatum*, ON138792 as *H. anatolicum anatolicum*, and ON146287 as *H. dromedarii*. There was no result obtained from *H. marginatum* confirming its morphological identity.

## Molecular results of *Theileria* from sheep and ticks

PCR amplification of *Theileria* spp. DNA was detected in 33 samples of sheep investigated while none of the goat samples revealed any amplification. A fragment of ~520 bp was amplified successfully from the positive samples. Therefore, the prevalence of *Theileria* spp. based on PCR results revealed that 33 (18.6%) of the sheep samples screened showed positive amplification of the partial 18S rRNA region. Twenty (26%) out of 77 blood samples were from Madina, whereas, 13 (13%) out of 100 of the blood samples were from Tabuk (Table 1). There was a significant difference in the prevalence based on the PCR results being higher in Madina ($X^2$ = 4.8277, $p$ < 0.05). The sensitivity of the PCR method compared to the direct microscopy was found to be 72% in the detection of *Theileria* sp. in the present study.

There was no significant difference in the prevalence of *Theileria* spp. concerning sex and rearing system, however, there was a significant difference in the prevalence of *Theileria* spp. in sheep less than 2 years old with those more than 2 years old using the PCR method ($X^2$ = 13.5767, $P$ < 0.05). Likewise, all positive samples were reported during the summer and none collected during the winter was positive.

Twenty DNA sequences were reported from sheep and were deposited at GenBank with the accession numbers ON138720–ON138721 and ON138724 to ON138741. Two additional sequences were obtained from ticks and deposited at GenBank with the accession numbers ON138722 and ON138723.

Positive amplification was obtained from 4/33 (12%) of ticks investigated from sheep and none yielded any PCR product from goats. Three of the positive ticks were from Madina 3/31(9.6%), while the fourth one was from Tabuk 1/32 (3%). The positive results were found in four adult ewes: three *H. impeltatum* ticks from Madina and one *H. dromedarii* from Tabuk. No *Theileria* spp. DNA was amplifiable from both *H. a. anatolicum* and *H. marginatum*. DNA sequencing of the PCR products obtained from ticks collected from sheep in Madina (*H. anatolicum*) and Tabuk (*H. dromedarii*) revealed that both harbored DNA identical to that of *Theileria ovis* based on the partial 18S rRNA region amplification.

## Phylogenetic analysis

Sequencing of the amplified product showed DNA which is identical to *Theileria ovis*. Representative DNA sequences obtained from sequencing of the 18S rDNA resulted from the amplification of DNA from sheep and ticks using primers that amplify *Theileria* DNA was deposited at the GenBank database (ON138720–ON138741). The sequences obtained were 99–100% similar to several other *Theileria ovis* deposited in GenBank. The DNA sequences from sheep and ticks in the present study grouped with DNA sequences from *T. ovis* from Saudi Arabia (Riyadh (MG738321), and Jeddah (MZ078465–MZ078473)), Turkey (MT883516 and MT883517), Iran (JN412662) and Hungary (KF681518). However, it was not grouped with *T. capreoli*, *T. cervi*, *T. luwenshuni*, and *T. mutans* and not with the transforming *Theileria; T. parva*, and *T. lestoquardi* (Fig. 3). Both trees generated by neighbor-joining and maximum likelihood revealed similar topology with distinct clade for *T. ovis* from different parts of the world and separate from other species

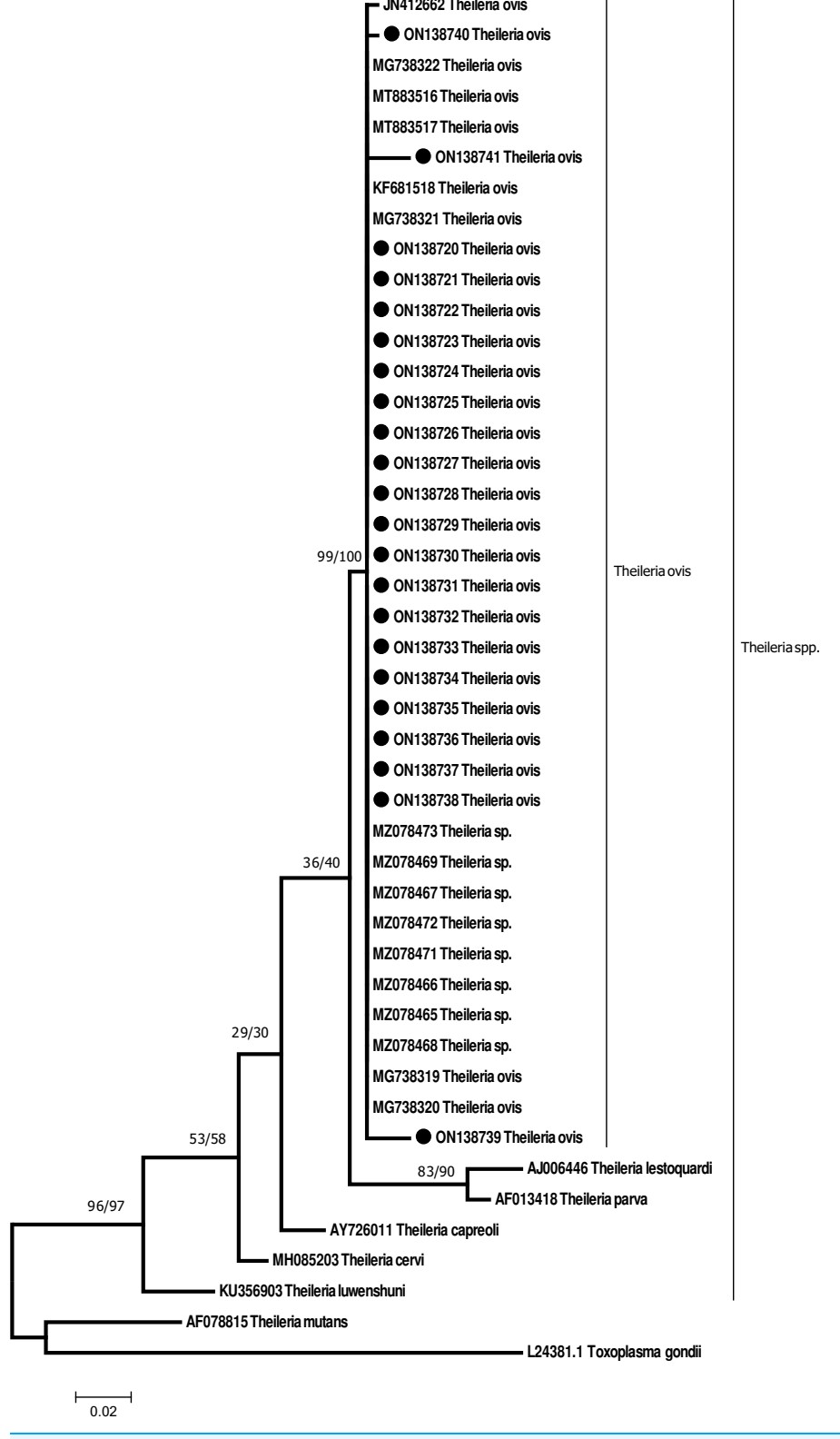

**Figure 3 A consensus phylogenetic tree constructed with maximum likelihood (ML) and neighbor-joining (NJ) methods, demonstrating phylogenetic relationships of *T. ovis* identified in the present study and related species of *T. ovis*, with *Toxoplasma gondii* as an outgroup inferred**

**Figure 3** (continued)
**from 18SrRNA gene sequence data generated from and other taxa from GenBank.** Numbers indicated at branch nodes are bootstrap values (ML/NJ). Samples given in solid circles are those reported from the present study.

of *Theileria* with strong bootstrap values. *Toxoplasma gondii* was used as an outgroup to root the tree.

18S rRNA sequences obtained from *Theileria* spp. in the present study were found to be 99–100% identical to sequences from *T. ovis* in the GenBank. There were 10 mutations in the alignment in three individuals (ON138739–ON138741). There were five mutations on sample no ON138739 from the sheep sample from Madina, one mutation on sample ON138740 from sheep from Tabuk, and four mutations on sample ON138741 from Tabuk as well. Regarding sample ON138738, the mutations were at positions; 77 A instead of C, 132 G instead of C, 293 G instead of T, 297 G instead of A, and at position 314 G instead of C. The mutation on samples ON138740 was at position 352 C instead of A. Mutations from sample ON138741 were at positions; 113 A instead of T, 205 A instead of G, 299 C instead of G, and at position 345 G instead of A.

## DISCUSSION

*Theileria ovis* reported in the present study from sheep in both Madina and Tabuk cities at a prevalence of 13.6% and 18.5% based on microscopic and molecular methods respectively. The prevalence was 18.2% and 10% using microscopic examination, whereas it was 26% and 13% using molecular techniques from Madina and Tabuk respectively. Goats examined from both cities were negative for *Theileria* spp. infection. Similar findings concerning the prevalence of *Theileria* spp. using both microscopy and molecular techniques were reported in previous studies (*Hajikolaei et al., 2003*; *Razmi, Hosseini & Aslani, 2003*; *Razmi, Eshrati & Rashtibaf, 2006*, *Durrani et al., 2011*; *Razmi & Yaghfoori, 2013*; *Zarei, Ganjali & Nabavi, 2019*; *Eliwa et al., 2021*). The higher prevalence of *Theileria* detected through molecular techniques compared to microscopic examination is likely due to the increased sensitivity of PCR, which can detect low levels of parasitemia that may not be visible microscopically. The possible explanation for the higher prevalence of *Theileria* spp. infection using DNA techniques compared to microscopic examination may be explained by the fact that the PCR is more sensitive and a single copy of the genome can be amplified. The microscopic method depends on the volume of the blood samples from which the smear was prepared; furthermore, if the level of parasitemia is low then lower chance of demonstrating the piroplasms in the blood smear. The sensitivity of the molecular method used in the present study was considered higher than previous studies on *Theileria annulata* in cattle (*Noaman, 2014*; *Ullah et al., 2021*). *Noaman (2014)* and *Ullah et al. (2021)* reported sensitivity of 57% and 69& respectively.

The partial *18S rRNA* gene sequence of *T. ovis* from the present study shared high identity (99–100%) with isolates from Saudi Arabia (Jeddah and Riyadh) and other countries including Turkey (MT883516 and MT883517), Iran (JN412662) and Hungary

(KF681518) (*Shemshad et al., 2012*; *Alanazi et al., 2019*; *Metwally et al., 2021*; *Kirman & Guven, 2023*). This may probably be due to the could be because of the movement of animals from one city to another. Which may help in the movement to ticks vectors responsible of transmitting *T. ovis* from one animal to another. Furthermore, the phylogenetic tree showed all *18S rRNA* sequences reported in the present study are in the same group with other *T. ovis* sequences available in GenBank, which confirm the identity of the organism recovered in this study.

T. ovis has been reported in Saudi Arabia since the early nineties of the last century by *Hussein et al. (1991)* from different regions in Saudi Arabia. In Tabuk region, it has been reported in 2009 by *Al-Khalifa et al. (2009)*. The prevalence of *Theileria* infection in sheep and goats has been reported in the Kingdom of Saudi Arabia as well as in different countries where the parasite is endemic. In most cases, the prevalence of sheep was far higher than the prevalence of goats which showed considerably lower prevalence or even absence of infection (*Hussein et al., 1991*; *El-Metenawy, 1999*; *Al-Khalifa et al., 2009*; *Gebrekidan et al., 2014*, *Azmi, Al-Jawabreh & Abdeen, 2019*). *Hussein et al. (1991)* reported, microscopically, a prevalence of *Theileria* spp. in 19.9% of sheep whereas they reported 6.9% in goats from the Qassim region, central Saudi Arabia. Similarly, from the same region *Theileria hirci* (now *T. lestoquardi*) was reported from 20.46% and 7.57% of sheep and goats respectively (*El-Metenawy, 1999*). Qassim region is an agricultural region which may provide suitable conditions for ticks to breed and infect animals whereas Madina conditions are almost desert conditions which may have some effect on tick breeding. Furthermore, both Madina and Tabuk are located at a higher altitude compare to Qassim and central Saudi Arabia.

*Alanazi et al. (2019)* reported a prevalence of *T. ovis* of 33.2% in sheep and 25.2% in goats using microscopic examination and a prevalence of 46% in sheep and 33.7% in goats using molecular techniques from Riyadh region in Saudi Arabia. A recent report from the western region of Saudi Arabia, particularly from Jeddah, revealed a high prevalence of *T. ovis*, using molecular techniques, as high as (57.8%, 48/83) from sheep and (51.9%, 112/216) from goats (*Metwally et al., 2021*). Azmi et al. (2019) reported *T. ovis* infection in Palestine from sheep and not from the goat samples they investigated, similar to what we have found in the present study. The high prevalence of *T. ovis* in both sheep and goats in western Saudi Arabia as a result of *Metwally et al. (2021)* compared to the results of the present study, may be attributed to the fact that exotic breeds were used in the study of *Metwally et al. (2021)* whereas only local breeds of sheep were investigated in the present study. Exotic breeds of ovine are highly susceptible to *Theileria* spp. compared to Indigenous and crossbreeds. Indigenous breeds are known to possess a natural ability to develop resistance to diseases transmitted by ticks when compared with exotic and crossbreeds (*Gebrekidan et al., 2014*; *Salih et al., 2007*).

In neighboring countries, such as Oman, a prevalence of *Theileria* spp. in sheep and goats was reported as 36.7% and 2.7% respectively using PCR amplification and sequencing of the partial 18S rRNA gene (*Al-Fahdi et al., 2017*). *T. ovis* was most prevalent in sheep (33.4%) while it was detected in 2.0% of goats. Similar findings were reported by *Arif et al. (2023)* where low prevalence of *T. ovis* was reported from goats compared to

sheep where higher prevalence was reported. The pathogenic *Theileria lestoquardi* was less prevalent and it was reported in (22.0%) of sheep and (0.5%) of goats (*Al-Fahdi et al., 2017*). In different studies, goats were found less susceptible to *Theileria* infection or negative for infection similar to what we have encountered in the present study (*Aktaş, Dumanli & Altay, 2005*; *Altay, Aktas & Dumanli, 2007*; *Gebrekidan et al., 2014*; *Fatima et al., 2015*). This finding was attributed to the nature of goat skin which is more resistant to tick attachment, unlike sheep skin which was found to be more suitable for tick attachment (*Fatima et al., 2015*). Another possible explanation is that the sporozoites of *T. parva* are not able to easily invade caprine lymphocytes which was hypothesized by *Syfrig et al. (1998)*. Whether this applies to *T. ovis* to goats in the Tabuk and Madina regions is unknown, therefore, further work is required to prove this hypothesis or reject it. It may also have resulted from either the absence of suitable vector-transferring *T. ovis* in goats or a difference in susceptibility to different breeds of goats.

The prevalence of *T. ovis* infection in females is greater than in males, however, the difference was no significant difference in the prevalence between males and females. Several investigators reported similar results (*Inci et al., 2008*; *Tuli et al., 2015*; *Ziam et al., 2020*; *Sray, 2021*; *Valente et al., 2023*). This may be explained by the fact that females have increased hormonal stress associated with pregnancy, childbirth, and feeding offspring (*Tuli et al., 2015*; *Sray, 2021*). It may be related to the fact that are kept for longer periods for breeding and rearing purposes. They may also receive less food for maintenance, unlike males who are fed to increase their body weight to meet their commercial needs (*Kamani et al., 2010*; *Parveen et al., 2021*).

*T. ovis* infection in sheep in both cities was significantly higher in animals less than 2 years compared to animals more than 2 years ($p < 0.05$). Similar findings were reported from other studies (*Naz et al., 2012*; *Yu et al., 2018*). This may be attributed to an underdeveloped immune system, deficient grooming, or another risk factor such as vaccination or weaning. Previously, it has been reported that goats 3–6 months of age had a higher risk of infection than other age groups (*Yu et al., 2018*). It is generally believed that adult animals are more prone to be infected with *Theileria* more than young animals unlike to what we have noticed in the present study. Previous studies have shown that *Theileria* infection in adult animals is greater than in young animals (*Aydin, Aktas & Dumanli, 2013*; *M'ghirbi et al., 2013*; *Metwally et al., 2021*). Their findings were attributed to the fact that ticks are associated with adult animals for longer periods compared to young animals hence higher chances of infection.

*Theileria ovis* infections in sheep in both cities studied were reported only from sheep collected during the summer and not during the winter. The same observations were recorded by *Durrani et al. (2012)* where they noted that the seasonal changes affect the spread of *Theileria* in sheep. This agrees with the findings of other studies, although the majority of the available data refer to adult ticks' activity (*Santos-Silva et al., 2011*; *Aydın, Vatansever & Arslan, 2022*). The increase in prevalence during hot seasons could be attributed to a higher tick infestation rate which is influenced by temperature, rainfall, and relative humidity (*Gosh et al., 2007*). Climatic conditions dictate the dynamics of tick-borne diseases by affecting tick distribution and seasonal occurrence (*Ahmed et al.,*

*2002*). Further work is required to identify *T. ovis* stages in different developmental stages of such ticks.

Potential vectors of *T. ovis* were investigated during the present study. Adult ticks identified from sheep and goats from Madina and Tabuk regions included; *H. dromedarii*, *H. impeltatum*, *H. anatolicum*, and *H. marginatum*. DNA of *T. ovis* was only amplified from *H. impeltatum* and *H. dromedarii*. In a previous study, *Omer, Alsuwaid & Mohammed (2021)* were able to identify *H. anatolicum* and *H. dromedarii* as vectors for *T. annulata* in cattle in the eastern province of Saudi Arabia. *H. anatolicum* as well as *H. marginatum* are probably not suitable vectors for transmitting *T. ovis* in sheep and goats in Madina and Tabuk. However, *H. dromedarii* is considered a suitable vector for both *T. annulalta* in cattle and *T. ovis* in sheep as it appeared in the present study as well as *Omer, Alsuwaid & Mohammed (2021)* work. Furthermore, nymphs of *H. dromedarii* have previously been demonstrated to transmit *T. annulata* experimentally in Sudan (*Mustafa, Jongejan & Morzaria, 1983*). In the present study, only adult ticks were identified and both *H. impeltatum* and *H. dromedarii* were named as potential vector which may transmit infection with *T. ovis* in sheep in both Madina and Tabuk. Further work is required to identify *T. ovis* stages in different developmental stages of such ticks.

Interestingly, *T. ovis* is considered to cause sub-clinical infection in small ruminants in contrast to the virulent *T. lestoquardi* (*Razmi & Yaghfoori, 2013*; *Yaghfoori et al., 2016*; *Yaghfoori, Mohri & Razmi, 2017*). In the present study, 10 mutations were detected in the 18S rRNA region studied. It is unknown whether these mutations may result in mild or severe sub-clinical infections in some sheep in the area or not. The detection of several mutations in the *18S rRNA* gene in this study, raises questions about potential variations in virulence, warranting further investigation. Further work is also needed concerning the pathogenicity of *T. ovis* in sheep.

## CONCLUSIONS

The prevalence of *Theileria* spp. in sheep and goats in Madina and Tabuk was investigated using microscopic and molecular methods. Only sheep was found to be infected with *Theileria ovis*, while goats were negative for infection. The prevalence of infection in sheep in Madina and Tabuk was fond to be 6% using microscopic methods. Likewise, only sheep yielded positive results on amplification of partial 18S rRNA gene, where 33 (18.6%) of the samples amplified successfully. DNA sequencing confirmed the identity of the parasite detected as *T. ovis*. Sheep from both cities are infected with *T. ovis* with significant difference in prevalence in young animals compared to older animals. The vectors responsible for the transmission of the infection were *Hyalomma dromedarii* and *Hyalomma impeltatum* where amplifiable DNA was reported from both ticks.

### Funding

This study was supported by the Researchers Supporting Project (RSP2024R25), King Saud University, Riyadh, Saudi Arabia; and also supported by Princess Nourah bint

Abdulrahman University Researchers Supporting Project number (PNURSP2024R437), Princess Nourah bint Abdulrahman University, Riyadh, Saudi Arabia. The funders had no role in study design, data collection and analysis, decision to publish, or preparation of the manuscript.

### Grant Disclosures
The following grant information was disclosed by the authors:
Researchers Supporting Project: RSP2024R25.
King Saud University, Riyadh, Saudi Arabia.
Princess Nourah bint Abdulrahman University Researchers Supporting Project: PNURSP2024R437.
Princess Nourah bint Abdulrahman University, Riyadh, Saudi Arabia.

### Competing Interests
The authors declare that they have no competing interests.

### Author Contributions
- Ruoa S. Almahallawi conceived and designed the experiments, performed the experiments, analyzed the data, prepared figures and/or tables, authored or reviewed drafts of the article, and approved the final draft.
- Sawsan A. Omer conceived and designed the experiments, analyzed the data, authored or reviewed drafts of the article, and approved the final draft.
- Esam M. Al-Shaebi performed the experiments, prepared figures and/or tables, authored or reviewed drafts of the article, and approved the final draft.
- Nawal Al-Hoshani analyzed the data, prepared figures and/or tables, authored or reviewed drafts of the article, and approved the final draft.
- Esam S. Al-Malki analyzed the data, prepared figures and/or tables, authored or reviewed drafts of the article, and approved the final draft.
- Rewaida Abdel-Gaber analyzed the data, authored or reviewed drafts of the article, and approved the final draft.
- Osama B. Mohammed conceived and designed the experiments, performed the experiments, analyzed the data, prepared figures and/or tables, authored or reviewed drafts of the article, and approved the final draft.

### Animal Ethics
The following information was supplied relating to ethical approvals (*i.e.*, approving body and any reference numbers):

The King Saud University ethics approval to carry out the study with the Department of Zoology Facility (Ethics Reference: KSU-SE.23-104).

### Field Study Permissions
The following information was supplied relating to field study approvals (i.e., approving body and any reference numbers):

A letter from the Department of Zoology was sent to the Ministry of Environment, Water and Agriculture to allow the collection of samples from sheep and goats.

## Data Availability

The sequences obtained from Theileria ovis and releated Theileria used in the present investigation are available at GenBank: ON138720–ON138741, ON138769–ON138791, ON146287, ON138792.

## Supplemental Information

Supplemental information for this article can be found online at http://dx.doi.org/10.7717/peerj.18687#supplemental-information.

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
