# Peer review of "Prevalence of Theileria ovis in sheep and goats in northwestern Saudi Arabia with notes on potential vectors"

_PeerJ, doi:10.7717/peerj.18687_

## Round 0.1 · original submission · Minor Revisions

Please find three reviews to your manuscript attached.

Rev 1 and 3 are very specific about the contents that you need to clarify; the role of season on sampling, how you picked the sampling size, the identification of the life stage of the ticks and a discussion on how these factors may have impacted on your results.

Reviewer 2 provided a selection of literature that may guide you to rework the manuscript- you may use these or similar papers by other authors that you are aware of.

Reviewer 2 requested clarification on the methods, please insert details as requested.

In conclusion, please address the matters related to sampling (and the statistical analysis of the data), respond to the suggestions and questions related to phrasing/language matters and methods and resubmit.

Please note that the manuscript may be returned to the same reviewers for a re-review.

·

Basic reporting

NO COMMENTS

Experimental design

NO COMMENTS

Validity of the findings

NO COMMENTS

Additional comments

THE WORK HAS BEEN EVALUATED IN DETAIL AND IT IS THINKING THAT THE ARTICLE WILL BE MORE CLEAR AND UNDERSTANDABLE AFTER THE PARTS SPECIFIED IN THE TEXT ARE CORRECTED.
THE CORRECTIONS MADE ''REVIEW'' BOX SELECTED CAN BE READ EASILY.
THANK YOU

Reviewer 2 ·

Basic reporting

.

Experimental design

.

Validity of the findings

.

Additional comments

How the sample size 403 was decided? Any formula use for sample size calculation? Please cite
Why Madina and Tabuk was used as sampling sites?
In abstract please mention all experiments conducted as well as number and species of ticks collected.
Were the parasites confirmed by DNA sequencing? If yes, briefly add this detail in abstract. This section is not written in a standard format. Please rewrite the abstract.
What is the conclusion of the abstract?
In the introduction section justify the novelty of this study. How it is different from previously published data?

Line 139... Do all parasite positive samples were sequenced? Please explain
What was the model used for the phylogenetic analysis?
How the ticks were identified? collected and preserved for analysis?
Please cite the reference for accession numbers used in the phylogenetic analysis, if they are published.
Please add a map showing location of the sampling site. briefly add geo climatic information of both areas and how they were different?
Please add Giemsa stain blood smear pictures showing parasites.
There are several papers published from Asia and middle east discussing the Theileria and Babesia in small ruminants. Please compare your results with them. Here are few examples.

PLOS ONE. 2024. 19(7): e0306697. https://doi.org/10.1371/journal.pone.0306697
PLOS ONE. 2023. 18(8), e0290620. https://doi.org/10.1371/journal.pone.0290620
PLOS ONE. 2023. 18(11): e0291302. https://doi.org/10.1371/journal.pone.0291302
Tropical Animal Health and Production. 53(4): 439 https://doi.org/10.1007/s11250-021-02870-5
. Pakistan Journal of Zoology. 2015. 47(2): 441-446.
Please compare the sensitivity of parasite detection methods by comparing smear screening and pcr results.

Reviewer 3 ·

Basic reporting

The article predominantly uses professional English, but some sections of the abstract and results could be worded more clearly. For example, the sentence: "Blood samples collected from 403 sheep and goats in Madina (n=201) or Tabuk (n=202)" could be rephrased for clarity. Some parts of the results may also benefit from simplified wording to reduce ambiguity, making the manuscript more accessible to a wider audience.

The manuscript reviews relevant studies in the field of Theileria and tick-borne diseases in sheep and goats. It provides adequate context for the study, showing its importance in light of previous research from Saudi Arabia and other regions. However, including more recent studies would strengthen the current understanding of Theileria epidemiology.

The manuscript follows a clear structure with sections for methods, results, and discussion. The figures and tables are appropriate and well-labeled, enhancing the presentation of the data. However, a few of the results in the tables could be more explicitly reported, particularly when comparing prevalence across different factors. The authors have shared the raw data and sequence information, ensuring transparency.

The study addresses its research question effectively, focusing on the prevalence of Theileria ovis in sheep and goats in northwest Saudi Arabia and identifying potential vectors. The results align with the original hypotheses, and the study remains focused throughout. However, the abstract lacks a brief mention of the phylogenetic analysis. Including this would provide a more complete overview of the findings.
The manuscript is well-structured and presents valuable findings about Theileria ovis in sheep and goats. Minor revisions are recommended to improve clarity in the abstract and results sections. Additionally, a brief mention of the phylogenetic analysis should be added to the abstract to give a fuller picture of the research.

Experimental design

The research aligns with the journal's focus on zoonotic diseases and epidemiological studies related to pathogens in livestock. It provides original data on the prevalence of Theileria ovis in sheep and goats in two cities in northwestern Saudi Arabia, an area with limited previous studies on this pathogen. The study fills a knowledge gap by identifying potential vectors for T. ovis transmission, which had not been previously reported in the region.

The research question is clearly stated and addresses a meaningful gap in the literature. The investigation seeks to determine the prevalence of Theileria in small ruminants and the potential role of ticks as vectors. This focus is relevant, given the impact of Theileria on livestock health and the economic significance of small ruminants in the region. The study contributes to understanding the epidemiology of tick-borne diseases in a previously understudied area, which is critical for developing effective control measures.

The study follows ethical guidelines for animal research, as indicated by the ethics approval from King Saud University. The sample size of 403 animals provides robust data, and the combination of microscopic and molecular techniques enhances the reliability of the findings. However, the investigation could benefit from a more detailed discussion on the limitations of the sample size and any potential biases that might affect the results, such as seasonal sampling.

The methods section is thorough, providing sufficient detail for replication. The description of sample collection, DNA extraction, and PCR amplification is clear, and the use of specific primers for Theileria detection is well justified. The study also includes details on tick species identification, molecular analysis, and phylogenetic tree construction, making the research replicable by other investigators. However, the authors could further clarify the statistical methods used to analyze prevalence data, particularly in relation to the different risk factors examined.

Validity of the findings

While the manuscript does not directly assess the broader impact and novelty of its findings, the study provides valuable epidemiological data on Theileria ovis in a region where such research is scarce. The authors focus on reporting prevalence rates and identifying vectors, which could be of significant interest to the scientific community. The paper presents a clear rationale for replicating similar studies in different geographic regions or in larger populations, emphasizing the benefit of expanding this research to enhance global understanding of Theileria infections in small ruminants.

The study is designed in a way that encourages replication. The rationale for the research—addressing gaps in knowledge about Theileria prevalence and its vectors in Saudi Arabia—is well articulated. Replication in other regions or with larger sample sizes could confirm these findings and further contribute to the literature. Additionally, the study's combination of microscopic and molecular techniques offers a robust framework for future research. Investigating seasonal variations and host factors in different contexts would help verify the conclusions drawn in this study.

The authors have shared the raw data, including PCR results and phylogenetic sequences, which ensures transparency and allows other researchers to verify the findings. The statistical analysis appears sound, with chi-square tests used to compare prevalence across different variables, such as age, sex, and rearing system. However, more detailed descriptions of potential confounding factors and how they were controlled would improve the robustness of the statistical interpretation. For example, additional control over seasonal variability and environmental factors affecting tick populations could be considered.

The conclusions are clear and appropriately linked to the original research question, which sought to determine the prevalence of Theileria ovis in sheep and goats and identify its vectors. The conclusions remain focused on the study's findings, avoiding overgeneralization. The authors accurately state that T. ovis was only found in sheep, not goats, and that Hyalomma dromedarii and H. impeltatum are likely vectors. These conclusions are firmly based on the results presented and do not extend beyond the scope of the data.

Additional comments

Description of the Work:

The article titled "Prevalence of Theileria ovis in sheep and goats in northwestern Saudi Arabia with notes on potential vectors" aims to assess the prevalence of Theileria species infection in small ruminants (sheep and goats) across two major cities in northwestern Saudi Arabia: Madina and Tabuk. A total of 403 blood samples were collected from both sheep and goats, analyzed using both microscopic and molecular techniques. The results indicated a 18.6% prevalence of Theileria in sheep, while no infection was found in goats. The study identified Hyalomma dromedarii and Hyalomma impeltatum as potential vectors of Theileria ovis in sheep. This is the first report providing data on the vectors involved in transmitting this pathogen in the region, contributing valuable epidemiological information on Theileria ovis.

Recommendation:

I recommend a minor revision of this manuscript, with a focus on improving the clarity of the abstract, expanding the results, and including a brief mention of the phylogenetic analysis, which is currently absent. Below are specific suggestions for revising the abstract.

Suggested Changes to the Abstract:

Original: "The prevalence of Theileria spp. infecting sheep and goats were investigated in two cities and surroundings in northwest Saudi Arabia." Change to: "The prevalence of Theileria spp. infecting sheep and goats was investigated in two cities and their surroundings in northwest Saudi Arabia."

Original: "Blood samples from 403 sheep and goats from Madina (n=201) and Tabuk (n=202) cities (177 from sheep and 226 from goats)." Change to: "A total of 403 blood samples were collected from sheep (n=177) and goats (n=226) in Madina (n=201) and Tabuk (n=202)."

Original: "The results indicated that: microscopic examination revealed 24 (6%) of the samples investigated showed intraerythrocytic bodies suggestive of Theileria." Change to: "Microscopic examination revealed intraerythrocytic bodies suggestive of Theileria in 24 (6%) of the samples."

Original: "A total of 33 (18.6%) sheep samples were positive for Theileria infection." Change to: "PCR analysis revealed that 33 (18.6%) of the sheep samples were positive for Theileria infection."

New Addition: "The phylogenetic analysis confirmed that the Theileria DNA sequences were identical to Theileria ovis, grouping closely with sequences from Saudi Arabia, Turkey, and Iran."

The authors should also present the results more explicitly, summarizing key findings such as the prevalence in different age groups, the absence of infection in goats, and the identification of vectors.

Introduction:
The introduction adequately sets the context for the study by emphasizing the significance of Theileria species in livestock and the economic importance of sheep and goats in Saudi Arabia. However, it could be enhanced by clearly articulating the existing knowledge gap and providing a stronger rationale for the research. For instance, the phrase "Sheep and goats may be infected by several pathogens with a wide variety of parasites among which the gastrointestinal parasitic infection is the most common" can be refined to "Sheep and goats are susceptible to a variety of pathogens, including gastrointestinal parasites, which are the most common; however, haemoparasitic infections, such as those caused by Theileria, also pose significant health risks." Additionally, while stating that "In Saudi Arabia, theileriosis in sheep was studied in the central part and a restricted part of the western region," it would be more accurate to say, "In Saudi Arabia, previous studies on theileriosis in sheep have primarily focused on the central and western regions, with limited data available from the northwest." Furthermore, the introduction should underscore the scarcity of information regarding the prevalence and impact of Theileria and potential tick vectors, emphasizing that despite the economic significance of small ruminants, data on haemoparasites in the region remains limited. A more explicit statement about the study's novelty and regional importance would effectively highlight how the findings contribute to filling this knowledge gap and to disease management strategies in livestock.

Discussion:
The discussion successfully interprets the findings and connects them to existing literature on Theileria prevalence and transmission. However, it could benefit from a greater emphasis on the significance of the results and a deeper exploration of their implications. For example, instead of stating that "The possible explanation for the higher prevalence of Theileria infection using DNA techniques compared to microscopic examination may be explained by the fact that the PCR is more sensitive," it would be clearer to say, "The higher prevalence of Theileria detected through molecular techniques compared to microscopic examination is likely due to the increased sensitivity of PCR, which can detect low levels of parasitemia that may not be visible microscopically." Moreover, the observation that "Theileria ovis infections in sheep in both cities studied were reported only from sheep collected during the summer and not during the winter" should be rephrased to highlight the implications of seasonality: "Theileria ovis infections were detected exclusively during the summer, suggesting a seasonal pattern in tick activity and pathogen transmission. This aligns with previous studies reporting a higher prevalence of Theileria during warmer months, likely due to increased tick infestation rates." Additionally, the discussion on Theileria ovis being associated with subclinical infections, unlike the more pathogenic T. lestoquardi, could include the note that "The detection of several mutations in the 18S rRNA gene in this study raises questions about potential variations in virulence, warranting further investigation." Finally, a more comprehensive analysis of the public health and economic implications of the findings would enhance the discussion. Expanding on the identification of vectors, specifically Hyalomma dromedarii and H. impeltatum, could provide insights into how this knowledge may inform future tick control strategies in the region.

---

## Round 0.2 · accepted · Accept

Dear Dr Mohammed

The reviewers are satisfied that you attended to the corrections . Reviewer 1 noticed a mistake related to a date in one reference (discrepancy between date in manuscript line 368 Aydin et al., 2013 and reference list which lists 2022) . Will you please attend to this corrections when you receive the proof?

·

Basic reporting

It is suitable

Experimental design

It is suitable

Validity of the findings

It is suitable

Additional comments

The article can be accepted after making the minor corrections indicated in the text. The corrections are included in the appendix.

Reviewer 3 ·

Basic reporting

No comment.

Experimental design

No comment.

Validity of the findings

No comment.

Additional comments

No comment.